nanotechnology/health and disease and epidemiology/microbiology

*in vitro*, MDR *E. coli*, graphene oxide-silver, photothermal treatment

**Authors for correspondence:**
Jun Ruan
e-mail: ruanjun818@sina.com
Yong Wang
e-mail: wy8486@sina.com

[†]These authors contributed equally to this study.

This article has been edited by the Royal Society of Chemistry, including the commissioning, peer review process and editorial aspects up to the point of acceptance.

# Photothermal-assisted antibacterial application of graphene oxide-Ag nanocomposites against clinically isolated multi-drug resistant *Escherichia coli*

Yuqing Chen[1,†], Wei Wu[2,†], Zeqiao Xu[3], Cheng Jiang[4], Shuang Han[5], Jun Ruan[3] and Yong Wang[3]

[1]Children's ENT Department, The Affiliated Wuxi Maternity and Child Health Care Hospital of Nanjing Medical University, 214122 Wuxi, People's Republic of China
[2]Cardiothoracic Surgery Department, The Affiliated Wuxi No. 2 People's Hospital of Nanjing Medical University, 214002 Wuxi, People's Republic of China
[3]Urology Surgery Department, and [4]Department of Laboratory Medicine, The Affiliated Wuxi People's Hospital of Nanjing Medical University, 214122 Wuxi, People's Republic of China
[5]School of Biotechnology, Jiangnan University, 214122 Wuxi, People's Republic of China

YW, 0000-0002-4244-852X

In the field of public health, treatment of multidrug-resistant (MDR) bacterial infection is a great challenge. Herein, we provide a solution to this problem with the use of graphene oxide-silver (GO-Ag) nanocomposites as antibacterial agent. Following established protocols, silver nanoparticles were grown on graphene oxide sheets. Then, a series of *in vitro* studies were conducted to validate the antibacterial efficiency of the GO-Ag nanocomposites against clinical MDR *Escherichia coli* (*E. coli*) strains. GO-Ag nanocomposites showed the highest antibacterial efficiency among tested antimicrobials (graphene oxide, silver nanoparticles, GO-Ag), and synergetic antibacterial effect was observed in GO-Ag nanocomposites treated group. Treatment with 14.0 µg ml$^{-1}$ GO-Ag could greatly inhibit bacteria growth; remaining bacteria viabilities were 4.4% and 4.1% for MDR-1 and MDR-2 *E. coli* bacteria, respectively. In addition, with assistance of photothermal effect, effective sterilization could be achieved using GO-Ag nanocomposites as low as 7.0 µg ml$^{-1}$. Fluorescence imaging and morphology characterization uncovered that bacteria integrity was disrupted after GO-Ag nanocomposites treatment. Cytotoxicity results of GO-Ag using human-derived cell lines (HEK 293T, Hep G2) suggested more than 80% viability remained at 7.0 µg ml$^{-1}$.

All the results proved that GO-Ag nanocomposites are efficient antibacterial agent against multidrug-resistant *E. coli*.

# 1. Introduction

Antibiotics have been widely used for over 70 years in different fields such as medicines, agriculture and environment [1]. However, bacteria have developed resistance to antibiotics through acquired resistance genes or intrinsic antibiotic resistance capability [2], which leads to the wide existence of resistant bacteria, including multidrug-resistant (MDR) bacteria, extensively drug-resistant (XDR) bacteria and pandrug-resistant bacteria [3]. In diagnosis practice, antibiotic resistance is a non-negligible concern for clinical physicians to treat infections such as post-operative infections, ear, nose and throat infections, and urinary tract infections. Antibiotic resistance makes antibiotic selection a great challenge as more and more MDR bacteria were found in patients [4–6]. Among all the infections encountered in hospital, urinary tract infections are one of the most common infections, in which case *E. coli* is found to be the major cause [7]. Moreover, spreading of certain type of antibiotic-resistant *E. coli* may become potential cause for epidemic disease [8]. Several scientific reports have indicated that antibiotic-resistant *E. coli* bacteria are found in soil, water, foods and animals [9–15]. Due to the wide spread of antibiotic-resistant bacteria and less efficiency of conventional antibiotics, alternatives are needed to deal with antibiotic-resistant bacteria.

Nanotechnology offers a new way to overcome healthcare challenges brought by antibiotic-resistant bacteria, as the nanoscale antimicrobial agents have huge surface to volume ratio, and the size is comparable to the pathogenic microbes, allowing it to penetrate into or contact with pathogenic microbes in an efficient way [16]. Most of the effective antimicrobials are metallic-based nanomaterials such as ZnO, TiO$_2$, silver and gold nanoparticles [17–20]. Among different nanomaterials, silver-based nanomaterials are the most popular antibacterial agent [21,22]. Silver nanoparticles (AgNPs) are effective against Gram-negative bacteria, Gram-positive bacteria, as well as MDR bacteria, while the antibacterial mechanism is still not clear though several hypotheses were propounded [23–26]. Graphene, a single layer of carbon atoms nanomaterial, has a unique set of physical, chemical and electronic properties [27–29], which enable it as novel agent for biomedical applications. Graphene-based nanomaterials are used as novel antimicrobial agents against broad-spectrum microbes via physical destruction of cell membrane and chemical damage brought by reactive oxygen species [30,31]. However, there are debates on antibacterial activity of graphene and graphene oxide (GO) as several papers claimed little antibacterial activity of graphene oxide [32,33]. Furthermore, graphene can serve as carrier for antibiotics delivery [34], and photothermal sensitizers in combination with near-infrared laser light for powerful photothermal killing of bacteria [35,36].

GO-Ag nanocomposites are widely reported to be efficient antimicrobial agents to different kinds of microbes, including standard bacteria, fungus and resistant bacteria [37–42]. It shows better antibacterial efficiency in comparison with silver nanoparticles, because GO sheet provides anchor platform to make AgNPs well dispersed and large contact surface between bacteria and AgNPs [40,43]. However, most of the studied bacteria were model bacteria instead of clinically isolated bacteria, especially MDR bacteria encountered in clinical practice. Furthermore, photothermal property of GO can be used for combined therapy to improve antibacterial efficiency, as well as to reduce risk of potential anti-nanomaterial resistance as silver-resistant bacteria were reported [44,45].

In order to find a way to address resistance problem of MDR bacteria in clinical practice, photothermal-assistant anti-MDR *E. coli* bacteria study is conducted using synthesized GO-Ag nanocomposites. To achieve reliable antibacterial efficiency, we break the study into four steps. First step is to verify the existence of synergistic effect of GO-Ag nanocomposite in comparison with AgNPs. Minimum inhibitory concentration (MIC) of AgNPs, GO-Ag and some antibiotics is used to evaluate the antibacterial efficiency according to standard protocol. Second step is to find possible cause for improved antibacterial efficiency of GO-Ag, and to clarify the debates on GO antibacterial activity. Antibacterial efficiency of GO, AgNPs, GO-Ag and a mixture of GO and AgNPs are studied simultaneously using a colorimetric assay with dye 3-(4,5-dimethyl-2-thiazolyl)-2,5-diphenyl-2H-tetrazolium bromide (MTT), namely MTT assay. In the third step, *in vitro* photothermal-assistant treatment of MDR bacteria is conducted with energy from near-infrared laser light to kill MDR bacteria in more efficient way. Finally, characterization of GO-Ag-treated bacteria is performed using fluorescence microscopy and scanning electron microscopy (SEM) to indicate resulting damage after treatment. Moreover, cytotoxicity of GO-Ag was tested using human-derived cell lines as it is a serious concern for any further applications.

# 2. Material and methods

## 2.1. Chemicals

$AgNO_3$, sodium citrate, expandable graphite flakes, sodium chloride, $H_2SO_4$, $KMnO_4$, 30% $H_2O_2$, HCl and NaOH are chemicals used for nanomaterials synthesis. Antibiotics including ampicillin, tetracyline, streptomycin and chloramphenicol are used for MIC testing. Tryptone, yeast extract, agar, dimethyl sulfoxide (DMSO) and MTT are used for bacteria culture and cell viability assay. All the reagents are purchased from Sigma-Aldrich.

## 2.2. Bacteria strain and cell lines

Two MDR *E. coli* strains are isolated from urine samples in clinical laboratory. These two bacteria are named MDR-1 and MDR-2, respectively, and their antibiotic susceptibility was tested in clinical laboratory. MDR-1 is resistant to penicillin (ampicillin, AMP), tetracyline (tetracyline, TET), quinolone (nalidixic acid, NAL) and aminoglycoside (streptomycin, STR). MDR-2 is resistant to penicillin (ampicillin, AMP), tetracyline (tetracyline, TET), quinolone (nalidixic acid, NAL), aminoglycosides (spectinomycin, SPE; gentamicin, GEN) and chloramphenicol (chloramphenicol, CHL).

Human embryonic kidney 293T cell (HEK 293T, ATCC CRL-1573) and human liver hepatocellular cell (Hep G2, ATCC HB-8065) were gifts from Jiangnan University.

## 2.3. Preparation and characterization of GO, AgNPs and GO-Ag nanocomposites

AgNPs were synthesized according to previously reported method [46]. First, 18 mg silver nitrate was added into 100 ml distilled (DI) water, and the solution was heated in oil bath until boiling. Afterwards, 20 mg sodium citrate in 2 ml DI water were added to boiled silver nitrate solution slowly. Then solution was kept boiling for 1 h. Synthesized AgNPs solution cooled to room temperature, and some sample were taken out for characterization. Graphene oxide sheets were prepared using a modified Hummer method [47,48]. Silver nanoparticles were deposited on GO sheets via reducing of silver nitrate using sodium citrate in GO aqueous solution. Briefly, 6 mg GO and 18 mg $AgNO_3$ were dissolved in 100 ml DI water under stirring in oil bath, 1% sodium citrate aqueous solution (2 ml) was added slowly in boiling solution. The solution was kept boiling for 1 h to produce GO-Ag nanocomposites. Then, the product was filtered and washed with DI water three times.

The content of silver in AgNPs and GO-Ag nanocomposites was measured using inductively coupled plasma atomic emission spectroscopy (ICP-AES). In detail, three samples were taken out and diluted at different level, then nitric acid was added to dissolve samples. Well dissolved samples were centrifuged to remove precipitate, supernatant samples were used for quantitative ICP-AES analysis. Concentrations of GO in GO-Ag nanocomposites were measured using spectrometer and weighing method after lyophilization. Extinction coefficient of GO was 21.2 mg ml$^{-1}$ cm$^{-1}$ at wavelength of 900 nm.

Absorption spectrum of GO (3 µg ml$^{-1}$), AgNP (4 µg ml$^{-1}$) and GO-Ag (7 µg ml$^{-1}$) dispersed in water were scanned using spectrophotometer. GO and GO-Ag were also characterized by transmission electron microscope (TEM; Tecnai G20 F20, FEI) and dynamic light scattering (DLS) and zeta potential (Zetasizer Nano, ZS90, Malvern). GO sheets were also characterized and analysed by atomic force microscope (AFM; Veeco) and software NanoScope 6.14R1. AgNP size distribution on GO-Ag nanocomposites was measured and analysed using software ImageJ 1.52d and JMP 14.2.0.

## 2.4. MIC of antimicrobial agents against MDR *E. coli*

According to a reported protocol [49], agar dilution method was used for MIC determination. At first, LB agar contained different concentrations of antimicrobials were prepared, and 2 µl bacteria suspension (10 000 CFU) were dropped on the surface of LB agar. LB agar plate was put upward for 0.5 h to allow inoculum fully absorbed into the agar. Afterwards, plates were incubated for 16 h in 37°C incubator. Two MDR bacteria strains susceptibility testing was done using a series of antimicrobials. They were GO-Ag, AgNPs, AMP, TET, STR and CHL, and corresponding concentrations were 0, 2, 4, 8, 16, 32, 64, 128, 256 and 512 µg ml$^{-1}$. Tests repeated at least three times.

## 2.5. *In vitro* antibacterial assessment of antibacterial agents

Single colony on agar plate was transferred to fresh LB broth, and then cultured for 12 to 16 h in 37°C shaker at speed of 250 r.p.m. After centrifugation, bacteria pellet was resuspended in 4.5 ml fresh LB broth to a final concentration of $10^7$–$10^8$ CFU ml$^{-1}$. Then, 0.5 ml nanomaterials were added into broth and mixed well. Afterwards, broth was incubated with nanomaterials at 37°C for 3 h before viability assessment. Tested antibacterial agents included GO, AgNPs, GO-Ag, GO + AgNPs (mixture of GO and AgNPs). Three levels of GO concentrations were 1.5, 3 and 6 µg ml$^{-1}$; silver concentrations were 2, 4 and 8 µg ml$^{-1}$; corresponding nanocomposites or mixture concentrations were 3.5, 7 and 14 µg ml$^{-1}$. Three samples were tested for each treated group.

Similar to reported protocol used to evaluate interaction between biomaterial and cell [50], MTT assay method was used to evaluate bacterial viability after incubation with antimicrobial agents. At first, 0.1 ml bacteria culture was transferred into 96-well plate, and mixed well with 20 µl 5 mg ml$^{-1}$ MTT solution for 16 h in 37°C incubator. After centrifugation (5 min, 2000 r.p.m.), supernatant was removed and replaced with 120 µl DMSO in each well. Afterwards, microplate was shaken for 10 min to dissolve precipitants. At last, OD570 were read by microplate reader (Bio-Rad). Three parallel samples were tested with three replicates for each sample.

## 2.6. *In vitro* photothermal treatment of MDR-2 *E. coli*

Overnight cultured MDR-2 bacteria were collected and resuspended in fresh LB broth to a final concentration of $10^7$–$10^8$ CFU ml$^{-1}$. Then, 10% volume of nanocomposites (GO, GO-Ag) or water (control) was added into LB broth, final concentration of GO and GO-Ag were 3 and 7 µg ml$^{-1}$. The broth was shaken at 37°C for 1 h before laser irradiation. Then, 1 ml bacteria culture was transferred into one well of 24-well plate, and 808 nm laser light was used to irradiate the broth. Irradiation lasted for 7 min at different energy density, including 0, 1.0, 1.5 and 2.0 W cm$^{-2}$. Temperature during laser irradiation was recorded by thermal camera. After irradiation, the mixture was cultured for 6 h before bacteria viability assessment. Viability assessment was conducted by spread plate method and MTT assay.

## 2.7. Fluorescence imaging of treated MDR-2 *E. coli*

Four groups of bacteria were prepared to observe their effects on survival status of bacteria, which were brought by photothermal irradiation (1.5 W cm$^{-2}$, 7 min) and GO-Ag treatment (7 µg ml$^{-1}$). After above treatment, MDR-2 *E. coli* were cultured for 6 h. Next, collected bacteria cells were washed and stained with 4′-6-diamidino-2-phenylindole (DAPI) and propidium iodide (PI). According to reported protocol [51], bacteria were washed in 10 mM MgSO$_4$ at pH 6.5 after centrifugation. And then, bacteria were stained with PI (5 µg ml$^{-1}$) and DAPI (5 µg ml$^{-1}$) for 30 min in dark. After drying in air, the samples were observed under confocal microscope (Leica TCS SP5 II).

## 2.8. Morphology characterization of GO-Ag-treated MDR-2 *E. coli*

MDR-2 *E. coli* with or without GO-Ag (14 µg ml$^{-1}$) was cultured in LB broth for 3 h. After that, the bacteria were collected and washed three times with phosphate-buffered saline (PBS, pH 7.4). Next, the bacteria were fixed with 2.5% glutaraldehyde solution, and then the cells were dehydrated by sequential treatment with 50%, 70%, 90% and 100% ethanol for 15 min. Finally, bacteria were transferred to a silicon wafer for gold sputter coating, and then imaged under SEM (Quanta 200FEG, FEI).

## 2.9. Mammalian cytotoxicity of GO-Ag nanocomposite

HEK 293T and HepG2 cells were cultured in Dulbecco's modified Eagle's medium (DMEM) supplemented with 10% fetal bovine serum. About 100 000 cells/well were seeded into 96-well microplates in the presence of 150 µl medium with GO-Ag nanocomposites, then cells were incubated for 24 h in an incubator containing 5% $CO_2$ at 37°C. Five replicates (wells) per concentration were conducted at different final concentrations of GO-Ag (0, 3.5, 7, 14, 20, 50 µg ml$^{-1}$). After 24 h incubation, 15 µl MTT solution (5 mg ml$^{-1}$) was added into microplates and well mixed before 4 h incubation. The microplates were then centrifuged at 2000 r.p.m. for 5 min. Supernatant was

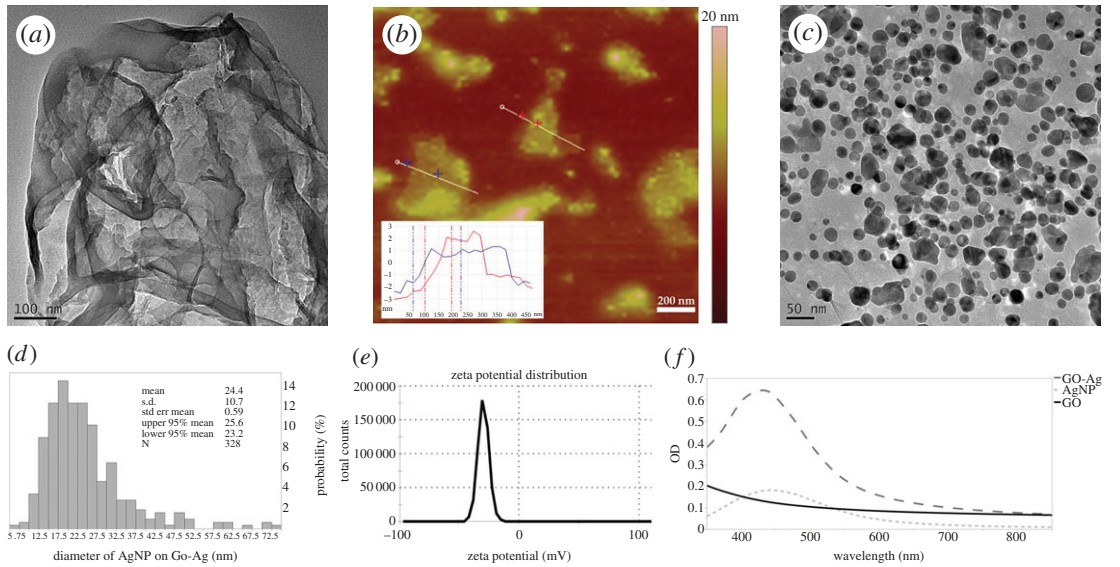

**Figure 1.** Characterization of synthesized nanomaterials. (*a*) TEM image of GO. (*b*) AFM image of GO. (*c*) TEM image of GO-Ag nanocomposites. (*d*) Size distribution of AgNP on GO-Ag. (*e*) Zeta potential distribution of GO-Ag. (*f*) Absorption spectrum of GO, AgNP and GO-Ag dispersed in water.

**Table 1.** Component content of GO-Ag and AgNP. Ag content was quantified using ICP-AES, and GO content was measured using spectrometer at 900 nm (extinction coefficient was 21.2 mg ml$^{-1}$ cm$^{-1}$). Dose = dilution factor × measured content.

| | | AgNP | GO-Ag | |
|---|---|---|---|---|
| sample | dilution factor | Ag content (µg ml$^{-1}$) | Ag content (µg ml$^{-1}$) | GO content (µg ml$^{-1}$) |
| no. 1 | 2 | 462 | 188 | 267 |
| no. 2 | 5 | 193 | 91 | 108 |
| no. 3 | 10 | 104 | 38 | 53 |
| mean dose (µg ml$^{-1}$) | | 976 | 404 | 535 |
| content (%) | | 100 | 43 | 57 |

discarded and replaced with 150 µl DMSO to dissolve purple crystals of formazan. At last, microplates were well mixed, and absorbance was measured at 570 nm by microplate reader (Bio-Rad).

# 3. Results

## 3.1. Antibacterial agents' characterization

In this study, GO, AgNP and GO-Ag were synthesized to investigate their antibacterial efficiency. Following reported protocol, GO sheets were prepared at first. As figure 1*a* shows, thin GO sheet was synthesized successfully. AFM analysis (figure 1*b*) revealed both single layer and multilayer sheets existed in synthesized GO. AgNPs were grown on GO sheets by *in situ* reducing silver nitrate solution on GO sheet. The synthesized GO-Ag nanocomposite was confirmed by observed absorption band around 440 nm (figure 1*f*) and TEM image (figure 1*c*). Figure 1*d* shows AgNP size distribution on GO-Ag; more than 90% AgNPs were in the range of 10–40 nm, and average diameter was 24.38 ± 10.74 nm. After silver nanoparticles decorated on GO sheets, zeta potential increased from −36.1 ± 6.98 mV (GO, electronic supplementary material, figure S1) to −28.8 ± 4.51 mV (GO-Ag, figure 1*e*). Size distribution of GO-Ag was characterized by DLS method; results were 147.4 ± 85.16 nm (electronic supplementary material, figure S2).

In order to quantify silver content in AgNP and GO-Ag, both materials were decomposed with nitric acid. Three levels of diluted samples were tested using ICP-AES, and dose was averaged results of three level samples. GO content in GO-Ag was measured using spectrometer at 900 nm. Results are shown in table 1;

**Table 2.** MICs of MDR *E. coli*.

| | GO-Ag (µg ml⁻¹) | AgNP (µg ml⁻¹) | AMP (µg ml⁻¹) | STR/CHL (µg ml⁻¹) | TET (µg ml⁻¹) |
|---|---|---|---|---|---|
| MDR-1[a] | 4 | 32 | >512 | >512 (STR) | 256 |
| MDR-2[a] | 4 | 32 | >512 | >512 (CHL) | 512 |

[a]MDR-1 and MDR-2 were two isolates of MDR *E. coli* from clinical samples. MDR-1 was resistant to ampicillin (AMP), tetracyline (TET), nalidixic acid (NAL) and streptomycin (STR), MDR-2 was resistant to ampicillin (AMP), tetracyline (TET), nalidixic acid (NAL) and chloramphenicol (CHL), spectinomycin (SPE) and gentamicin (GEN).

synthesized AgNP dose was 976 µg ml⁻¹, and averaged silver and GO content in GO-Ag nanocomposites was 43% and 57%, respectively.

## 3.2. MIC testing using agar dilution method

MIC is defined as the lowest concentration of antimicrobial agent that inhibits visible growth of microorganisms after overnight incubation. AgNP, GO-Ag and antibiotics (penicillin, tetracyline and aminoglycoside) were tested in two clinical MDR bacteria strains. Results (table 2) showed that both bacteria were resistant to AMP, TET, STR or CHL. MIC dose for all typical antibiotics are greater than or equal to 256 µg ml⁻¹, while MIC dose of nanomaterial-based antibacterial agents are much lower. As shown in table 2, MIC dose of GO-Ag and AgNPs were 4 and 32 µg ml⁻¹, respectively. Compared with AgNP, a widely used antimicrobial agent, GO-Ag showed higher antibacterial efficiency against two clinical MDR *E. coli* strains; 4 µg ml⁻¹ GO-Ag could completely inhibit growth of MDR-1 and MDR-2 bacteria on LB agar. No visual colony was observed on LB agar with 10 000 CFU bacteria cultured on agar media overnight.

## 3.3. Synergetic antibacterial effect verification of GO-Ag

In order to find why GO-Ag nanocomposites showed higher antibacterial efficiency than AgNP, we decomposed GO-Ag nanocomposites and measured the ratio of GO and Ag by ICP-AES and spectrometer. Results (table 1) showed content of GO and AgNP were 57% and 43% respectively in GO-Ag nanocomposite. Afterwards, three levels of AgNP (2, 4, 8 µg ml⁻¹), GO (1.5, 3, 6 µg ml⁻¹) and GO-Ag (3.5, 7, 14 µg ml⁻¹) nanocomposites were tested for antibacterial efficiency comparison. Besides, GO + AgNP mixture was used as control group to verify existence of synergetic effect of GO-Ag nanocomposites.

Antibacterial efficiency testing was studied in LB broth, and 3 h incubation time was selected as bacteria were in the middle of log phase. Thus, it is easier for us to discriminate minor difference between treated and control groups.

Quantification results are shown in figure 2a,b. GO-Ag showed the highest antibacterial efficiency among tested nanomaterials. As concentration increased, remaining viabilities of two MDR *E. coli* strains decreased obviously in GO-Ag-treated group. Correspondingly, remaining viabilities of MDR-1 and MDR-2 were 77.1%, 52.2%, 4.4% and 88.9%, 66.8%, 4.1%, respectively. On the other hand, there was no obvious viability decrease trend with increased concentrations of GO or AgNP. In AgNP-treated group, the remained viabilities of MDR-1 and MDR-2 were in the range of 81.8%–99.1%. In GO-treated group, the remained viability of MDR-1 and MDR-2 were in the ranges of 87.7% to 95.1% and 99.3% to 105.9%. Compared with GO-Ag nanocomposite, GO + AgNP mixture did not show observable viability decrease trend with increasing mixture concentration. The above results demonstrated that high antibacterial efficiency of GO-Ag arose from synergetic effect of GO and AgNPs after they combined as composite. Moreover, two MDR *E. coli* bacteria showed similar response to different nanomaterial. A minor difference was that MDR-2 bacteria were less susceptible to GO-Ag than MDR-1 bacteria at low concentration. When GO-Ag concentration was 3 µg ml⁻¹ no significant inhibition was observed in MDR-2 *E. coli* group.

## 3.4. *In vitro* photothermal-assisted treatment of MDR-2

Photothermal treatment of MDR-2 *E. coli* using GO-Ag nanocomposites was conducted. MDR-2 was selected as it was non-susceptible to more antibiotics and less susceptible to GO-Ag at lower concentration (figure 2b).

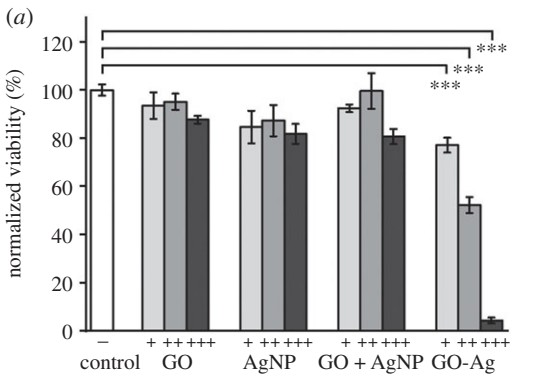
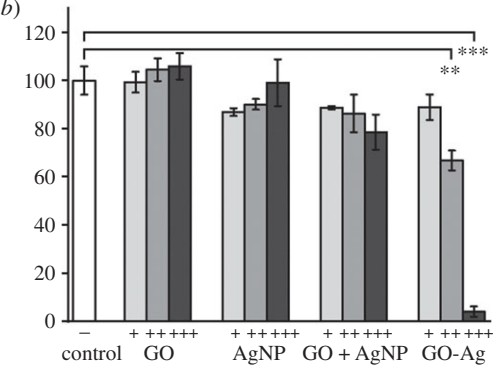

**Figure 2.** MDR *E. coli* viability assessment by MTT assay. (*a*) MDR-1 *E. coli* viability and (*b*) MDR-2 *E. coli* viability after treatment with GO, AgNP, GO-Ag, mixture of GO and AgNP (GO + AgNP) at different final concentrations in LB broth. +, ++, +++ represent different concentrations for different groups. GO (+, 1.5 µg ml$^{-1}$; ++, 3 µg ml$^{-1}$; +++, 6 µg ml$^{-1}$); AgNP (+, 2 µg ml$^{-1}$; ++, 4 µg ml$^{-1}$; +++, 8 µg ml$^{-1}$); GO+AgNP mixture (+, 1.5 µg ml$^{-1}$ GO and 2 µg ml$^{-1}$ AgNP; ++, 3 µg ml$^{-1}$ GO and 4 µg ml$^{-1}$ AgNP; +++, 6 µg ml$^{-1}$ GO and 8 µg ml$^{-1}$ AgNP), GO-Ag (+, 3.5 µg ml$^{-1}$; ++, 7 µg ml$^{-1}$; +++, 14 µg ml$^{-1}$). Error bars represent standard deviations ($n \geq 3$). $^{*}p < 0.05$, $^{**}p < 0.01$, $^{***}p < 0.001$.

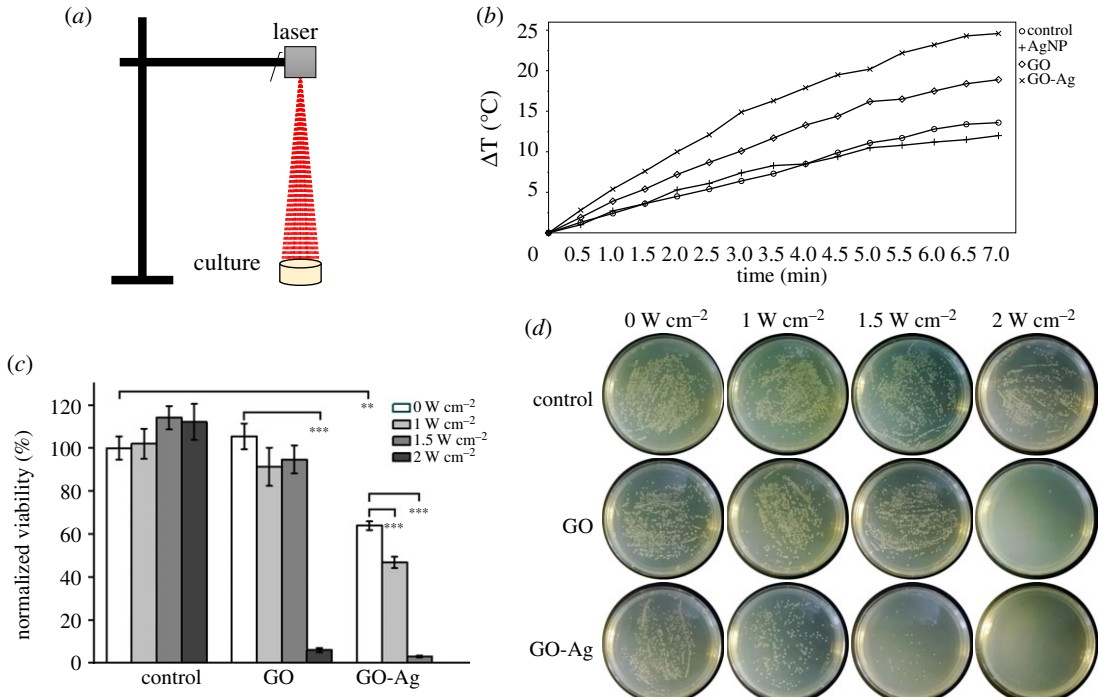

**Figure 3.** *In vitro* photothermal treatment of MDR-2 *E. coli*. (*a*) Illustration of photothermal treatment of bacteria in LB broth. (*b*) Heating curves of nanomaterials (7 µg ml$^{-1}$ GO-Ag, 4 µg ml$^{-1}$ GO, 3 µg ml$^{-1}$ AgNP) dispersed in LB broth, 1.5 W cm$^{-2}$. (*c*) Remaining bacteria viability after photothermal treatment, 3 µg ml$^{-1}$ GO, 7 µg ml$^{-1}$ GO-Ag. (*d*) Photos of bacteria colonies on LB agar plates after photothermal treatment. Error bars represent standard deviations ($n \geq 3$). $^{*}p < 0.05$, $^{**}p < 0.01$, $^{***}p < 0.001$.

First, $10^7$–$10^8$ CFU ml$^{-1}$ bacteria were cultured with 7 µg ml$^{-1}$ GO-Ag nanocomposites or 3 µg ml$^{-1}$ GO in LB broth for 1 h. Afterwards, 808 nm near-infrared (NIR) laser irradiation was exerted continuously on bacteria culture for 7 min, as illustrated in figure 3*a*. After 6 h incubation, bacteria were collected for cell viability assessment using MTT assay and spread plate method. As shown in figure 3*c,d*, results of two viability assessment methods were consistent with each other. A total of 6 h incubation time was used because bacteria in control group were in plateau phase. In this case, we could figure out whether bacteria growth was stopped completely after photothermal treatment.

Before photothermal treatment, NIR heating curves of GO-Ag, GO, AgNP dispersed in LB broth were recorded by thermal camera. Temperature increasement of control, AgNP, GO and GO-Ag were 13.6°C, 12°C, 18.9°C and 24.6°C after 7 min irradiation at 1.5 W cm$^{-2}$ (figure 3*b*). As a matter of fact, GO-Ag owned property of converting electromagnetic energy to heat, which might elevate its antibacterial

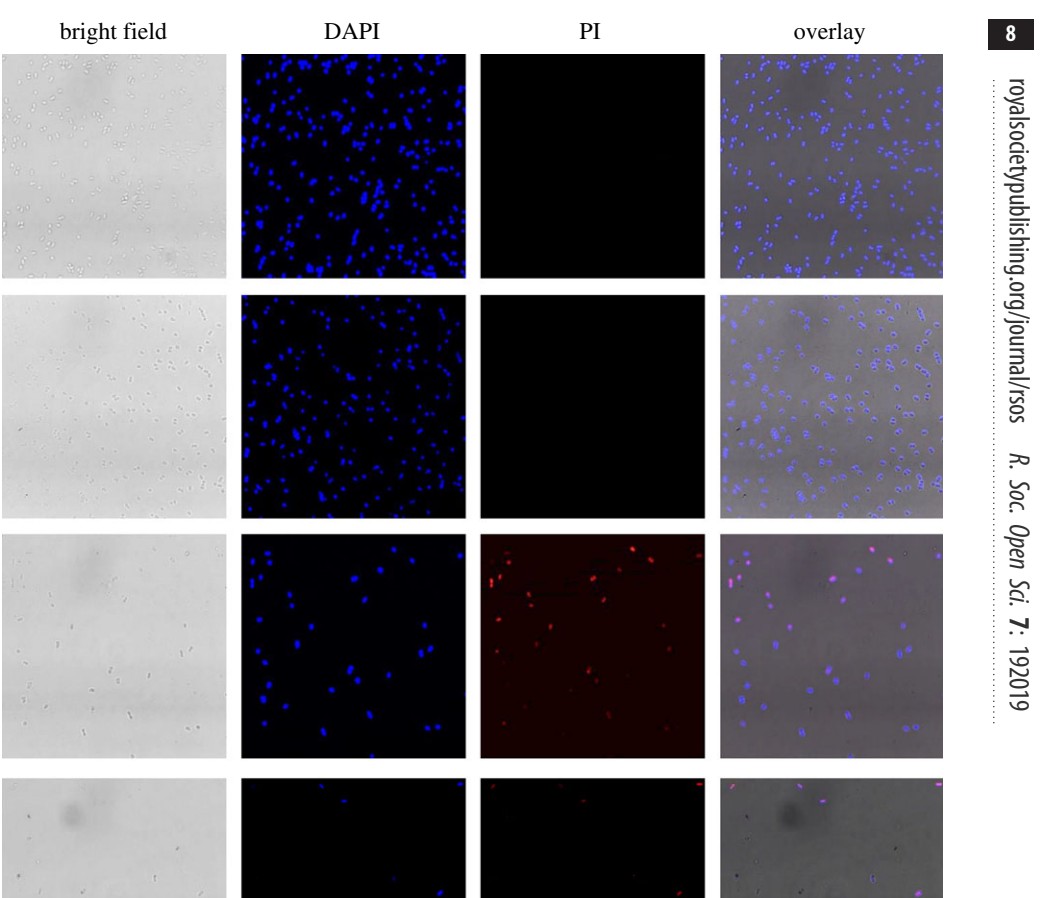

**Figure 4.** Fluorescence imaging of MDR-2 *E. coli* with and without photothermal treatment. DAPI and PI were used to stain the nuclei acid of bacteria. The difference is DAPI stains all the bacteria while PI can only stain the non-living bacteria. DAPI-stained bacteria emit blue fluorescence light while PI-stained bacteria emit red fluorescence light. 'Laser +' and 'Laser −' denote the presence and absence of photothermal treatment, 'GO-Ag +' and 'GO-Ag −' denote the presence and absence of GO-Ag nanocomposites. 1.5 W cm$^{-2}$, 7 µg ml$^{-1}$ GO-Ag. Scale bar is 5 µm.

efficiency. In figure 3*c*, different laser irradiation (1.0, 1.5 and 2.0 W cm$^{-2}$) had no observable inhibitory effect to MDR-2 *E. coli* in control groups. Thus, photothermal treatment comparison between GO and GO-Ag was suitable. Since GO content in 7 µg ml$^{-1}$ GO-Ag nanocomposite was 43%, 4 µg ml$^{-1}$ GO was used as control. In figure 3*c*, compared with GO sheets, GO-Ag was more efficient to resist the growth of bacteria at the same power density. This was because heating curves of GO-Ag were above GO at the same power density (electronic supplementary material, figure S3) and GO-Ag could inhibit growth of bacteria even without photothermal treatment. When power density increased to 2 W cm$^{-2}$, GO heating curve overlaid with 1.5 W cm$^{-2}$ GO-Ag, and remaining bacteria viabilities were similar for GO and GO-Ag in this condition. As shown in figure 3*c*, remaining viabilities were 3.0% in GO-Ag group at 1.5 W cm$^{-2}$, and 6% in GO-treated group at 2 W cm$^{-2}$. When power density was increased to 2 W cm$^{-2}$, MDR-2 *E. coli* could be killed completely in GO-Ag-treated group. However, the remaining viability was 69% without photothermal treatment in GO-Ag group. Thus, efficient antibacterial effect could be achieved with lower concentration of GO-Ag when photothermal treatment was exerted simultaneously.

## 3.5. Characterization of GO-Ag-treated MDR bacteria

The above results demonstrated that GO-Ag nanocomposites were highly effective antibacterial agent, but whether bacteria growth was inhibited temporarily or permanently remained unclear. Fluorescence imaging, a well-accepted qualitative method, was used to distinguish living and non-living bacteria in treated groups (red colour in figure 4) other than efficiency comparison (figure 3).

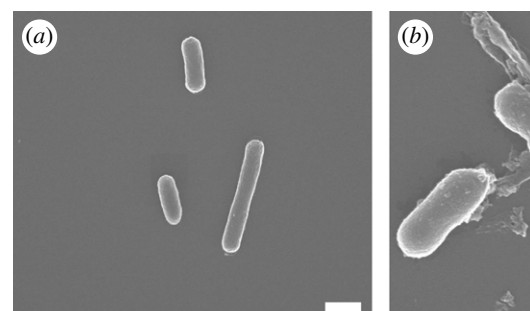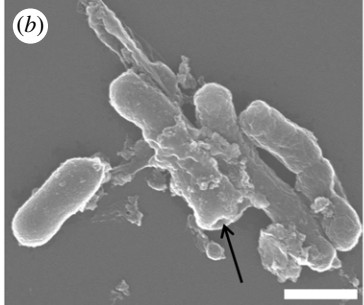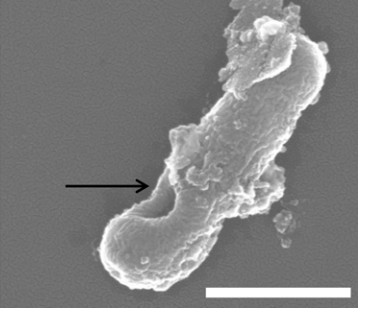

**Figure 5.** SEM images of MDR-2 *E. coli*. (*a*) SEM image of control group without GO-Ag treatment. (*b*) SEM images of bacteria treated with 14 µg ml$^{-1}$ GO-Ag. Arrows show disrupted cell wall. Scale bar is 1 µm.

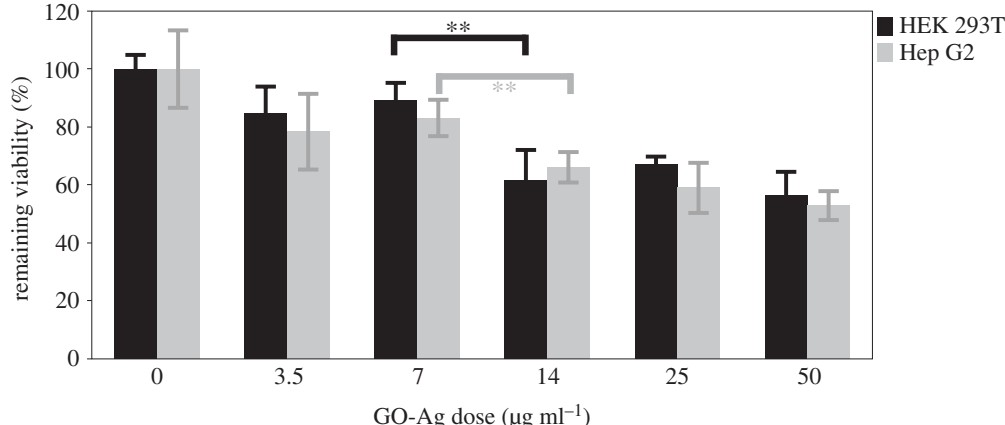

**Figure 6.** Cytotoxicity of GO-Ag nanocomposites. HEK 293T and Hep G2 cells viability assessment was conducted using MTT assay after 24 h incubation with GO-Ag nanocomposites. Error bars are standard deviations of five replicates. $^*p < 0.05$, $^{**}p < 0.01$, $^{***}p < 0.001$.

Specifically, compared with million to billion cells used in viability assessment in figure 3, fluorescence images were snapshots of extremely small parts of treated bacteria, which was not suitable for quantitative comparisons. In this experiment, photothermal irradiation and GO-Ag treatment were two variables that influence bacteria viability. Fluorescence imaging was conducted to observe changes of bacteria living status that was induced by these two variables. As shown in figure 4, four groups of bacteria were imaged using fluorescence microscopy. In the two control groups without GO-Ag nanocomposites addition (GO-Ag –), dead cells could hardly be found under microscope and the laser irradiation did not induce bacteria death, which was consistent with results from figure 3*b*. By contrast, in the group with GO-Ag nanocomposites (GO-Ag +), most of the bacteria were non-living cells. Moreover, in the group with GO-Ag nanocomposites and laser irradiation (GO-Ag +, laser +), fewer cells could be found in comparison with group without laser irradiation (GO-Ag +, laser –). Results indicated that both GO-Ag treatment and photothermal treatment could result in non-living cells.

Furthermore, morphology of MDR-2 bacteria was characterized using SEM after treatment with 14 µg ml$^{-1}$ GO-Ag nanocomposites. As shown by arrows in figure 5, some pits were found on the surface of cell wall, which meant cell walls were disrupted after GO-Ag nanocomposites treatment. While in the control group, cell wall was smooth, and no cell integrity disruption was observed.

## 3.6. Cytotoxicity of GO-Ag nanocomposites

Given wide concerns about cytotoxicity of nanomaterials, cytotoxicity of GO-Ag was conducted using HEK 293T and Hep G2 cells. These two cell lines are human origin cells, HEK 293T is a highly transactable derivative of human embryonic kidney 293 cells and Hep G2 is human hepatoma-derived cell line. Results in figure 6 show that cytotoxicity was correlated with dose of GO-Ag; high dose GO-Ag would cause a significant decrease in cell viability. As shown in figure 6, remaining viabilities of HEK 293T and Hep G2 at 14 µg ml$^{-1}$ were about 61.7 ± 10.4%, 66.1 ± 5.3% and 56.5 ± 8.0%, 52.9 ± 5.0%

at 50 µg ml$^{-1}$. However, remaining viabilities of both cell lines were above 80% when concentration reduced to 7.5 µg ml$^{-1}$, and MDR-2 *E. coli* can be completely killed with assistance of photothermal treatment at 7.5 µg ml$^{-1}$. Which suggested that GO-Ag mediated photothermal combined treatment was a promising method with low cytotoxicity at 7.5 µg ml$^{-1}$.

# 4. Discussion

It was reported GO-Ag could not only inhibit growth of non-susceptible bacteria, including Gram-positive *S. aureus* and Gram-negative *E. coli* [38,46], but also inhibit growth of methicillin-resistant *S. aureus* [39] and fungus like *Candida albicans* and *Candida tropical* [42]. In this work, clinically isolated MDR *E. coli* was selected as samples to test the antibacterial efficiency of GO-Ag nanocomposites *in vitro*. It was proved that GO-Ag nanocomposites were effective against MDR *E. coli*, which broadens its antimicrobial spectrum. Moreover, the most important advance was photothermal treatment was combined for the first time in GO-Ag antibacterial applications.

GO-Ag nanocomposites showed antibacterial effect in both liquid and solid media (table 2 and figure 2). As MIC results showed, 4 µg ml$^{-1}$ GO-Ag could completely inhibit growth of $10^4$ CFU bacteria on agar plate. In broth, ratio of survival bacteria was less than 5% when 14 µg ml$^{-1}$ GO-Ag was added into $10^7$–$10^8$ CFU ml$^{-1}$ bacteria culture. AgNPs also showed antibacterial effect, but efficiency was not so obvious. MIC of AgNP was 32 µg ml$^{-1}$ on agar medium. While 8 µg ml$^{-1}$ AgNP addition in broth had minor antibacterial effect on two MDR bacteria. GO-treated bacteria, showed no viability decrease trend with increased concentration from 1.5 to 6 µg ml$^{-1}$, though several published papers reported that GO was an antimicrobial agent. The different antibacterial property of GO might come from difference of material preparation method, dose, material size, culturing condition and sample handling method [47–52].

Higher antibacterial efficiency of GO-Ag nanocomposites can be explained by AgNPs being well distributed on GO sheets, which provided large contact area between bacteria and AgNPs. Compared with GO, AgNPs and mixture of GO and AgNPs, GO-Ag nanocomposite-treated group achieved better antibacterial result at same concentration of Ag or GO, which disclosed existence of synergetic effect that arose from combining GO and AgNPs as composite. And this synergetic effect was consistent with previous reported works [38,39].

Compared with other reported GO-Ag-related antimicrobial applications, photothermal therapy was combined in this study for the first time. An important benefit was that GO-Ag dose could be cut down, which was an advantage for biomedical applications; 7 µg ml$^{-1}$ GO-Ag could completely killed bacteria in photothermal combined therapy, while 14 µg ml$^{-1}$ GO-Ag could not completely inhibit MDR bacteria growth. Photothermal-assistant therapy could greatly enhance antibacterial efficiency as GO-Ag possessed advantages of AgNP and GO. AgNP inhibited bacteria growth, while GO sheets absorbed the NIR light and generated heat to help kill bacteria. Another advantage of photothermal therapy was that it could kill MDR bacteria even if bacteria were resistant to GO-Ag. Thus, photothermal-assistant therapy with GO-Ag provides an alternative strategy to solve problems brought by drug-resistant bacteria.

In order to better understand interaction between GO-Ag nanocomposites and bacteria, fluorescence imaging (figure 4) and SEM imaging (figure 5) of treated *E. coli* were analysed. Fluorescence imaging was used to distinguish living and dead bacteria using optimized PI staining process for bacteria [53]. DAPI was used to stain DNA-containing bacteria regardless of their physiological status, while PI was used to stain membrane-compromised bacteria [54–56]. In fluorescence images, red light emitted from GO-Ag and photothermal-treated groups demonstrated that both GO-Ag nanocomposites and photothermal treatment could damage bacteria integrity. Morphology characterization using SEM confirmed that cell integrity was disrupted by GO-Ag nanocomposites, which was consistent with published results [38,39,46].

Toxicity of engineered nanomaterials is an important consideration for further applications, especially for newly synthesized nanomaterials. Toxicity of GO and AgNPs is widely explored both *in vivo* and *in vitro* [57–61]. Cytotoxicity of GO-Ag nanocomposites was also reported by several researchers. De Luna *et al.* had done comparative toxicity study using GO-Ag and pristine counterparts (GO, AgNP), and viability IC$_{50}$ value of macrophage cells (J774, peritoneal macrophages) derived from murine were compared. Results showed IC$_{50}$ of GO, Ag, GO-Ag were 16.9, 8.9 and 2.9 µg ml$^{-1}$ in macrophage J774 cell after 24 h exposure [62]. Tang *et al.* reported 80% cell viability remained after 24 h incubation of mammalian cells (HEK 293T, HeLa) and 10 µg ml$^{-1}$ GO-Ag [38]. In our study, cytotoxicity of GO-Ag was assessed using human-derived liver and kidney cell lines. Results showed

more than 80% viability remained for both cell lines, which suggested our synthesized GO-Ag was a great antibacterial agent with low cytotoxicity.

This study provided a facial method to synthesize GO-Ag nanocomposite, which possessed synergetic antibacterial property as well as photothermal property. It provided double insurance to combat traditional antibiotic-resistant bacteria, and achieved enhanced antibacterial efficiency to pathogens with reduced cytotoxicity. Given the excellent antibacterial performance of GO-Ag nanocomposites against widely distributed MDR *E. coli* bacteria, they will be a useful antimicrobial in future medical applications.

# 5. Conclusion

Compared with widely used AgNP, GO-Ag nanocomposite showed much better antibacterial efficiency to clinically isolated MDR *E. coli*. Moreover, photothermal treatment could be combined to further lower the dose to 7 µg ml$^{-1}$, and MDR *E. coli* were killed completely with reduced cytotoxicity. Fluorescence imaging and morphology characterization disclosed that bacteria integrity was damaged after treatment with GO-Ag. Given the excellent antibacterial performance of GO-Ag nanocomposites against widely distributed MDR *E. coli* bacteria, they could be useful antibacterial consumables in future medical applications.

Data accessibility. Our data are deposited at Dryad Digital Repository: https://doi.org/10.5061/dryad.s7h44j133 [63].
Authors' contributions. Y.C. and W.W. contributed equally to this work, prepared all samples and performed experiments. Z.X. and C.J. collected and analysed the data. S.H. and Y.W. interpreted the results and wrote the manuscript. J.R., S.H. and Y.W. designed the research. All authors gave final approval for publication.
Competing interests. The authors declare no competing interests.
Funding. This work is supported by Young Project of Wuxi Health and Family Planning Commission (grant no. Q201809).
Acknowledgements. We thank Dr Wu for fruitful discussion.

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
