## [Reviewer comments · Royal Society Open Science]

Review History

RSOS-192019.R0 (Original submission)

Review form: Reviewer 1

Is the manuscript scientifically sound in its present form?

No

Are the interpretations and conclusions justified by the results?

No

Is the language acceptable?

No

Do you have any ethical concerns with this paper?

No

Have you any concerns about statistical analyses in this paper?

No

Recommendation?

Reject

Comments to the Author(s)

Remarks to Authors,

This manuscript describes a graphene oxide-silver (GO-Ag) nanocomposites as an antibacterial agent to treat multidrug-resistant (MDR) bacterial infection. Ag nanoparticles have an excellent antibacterial effect due to the toxicity of Ag ion. GO mediated photothermal effect will potentially cause bacterial death. Synergetic antibacterial effects can against clinical isolated MDR bacterial. However, some major defects make it difficult to accept.

1. The characterization of GO-Ag nanocomposites is not clear. The contents of each component are unknown, such as the ratio of GO and Ag in GO-Ag, respectively. It is not simply put these two together, which may cause wrong results. Such as the concentration of the three components Ag NP, GO, GO-Ag were 4 $\mu\text{g m/L}$, 3 $\mu\text{g m/L}$ and 7 $\mu\text{g m/L}$. And It also may cause some unmatched results such as in Figures 1c and 1d. The absorbance of GO-Ag at 808 nm is lower than the GO, however, the photothermal effect of GO-Ag is higher than the GO at the same concentration of GO. These results indicated that the concentration of GO-Ag may be wrong. The more detail quantitatively analysis of the GO-Ag is needed, such as through the TGA and ICP-MS.
2. In Figure 3b, the GO mediated photothermal effect shows a dramatically antibacterial efficiency at a power density of 2 W/cm². however, it cannot be observed at a power density of 1.5 W/cm². The results should be carefully checked again. And the antibacterial transition power density between 2 and 1.5 W/cm² should be provided to make it more convincing.
3. The results in Figure 4 (Laser- GO-Ag+ group) shows that less than 20 % of cells are alive, which are not consistent with the results in Figure 2 (at least 60 % cells alive at the same concentration of GO-Ag).
4. The English writing should be much improved.

Review form: Reviewer 2**Is the manuscript scientifically sound in its present form?**

Yes

Are the interpretations and conclusions justified by the results?

Yes

Is the language acceptable?

Yes

Do you have any ethical concerns with this paper?

No

Have you any concerns about statistical analyses in this paper?

No

Recommendation?

Accept with minor revision (please list in comments)

Comments to the Author(s)

The authors report antibacterial and photothermal applications of GO-Ag nanocomposite using clinical isolated bacteria as examples. Nevertheless, there are some concerns as below:

- 1) Characterization of GO-Ag / quality control

To reproducibly obtain the same results, proper physicochemical characterization of the GO-Ag nanocomposite should be done. E.g. what about the zeta-potential of the prepared GO-Ag nanocomposites, which actually affect their interaction with bacterial membrane? Is the GO single layer or multiple layers? What about the size distribution of Ag NP synthesized on GO?

2) Toxicity of GO-Ag

When the author talk about future medical applications in section "5.Discussion", they claim that "...GO-Ag nanocomposites can be coated onto consumables such as bandage, gauze and woundplast, or surgical devices...". However, there is a serious concern on the toxicity of Ag/Ag+ to human body? There should be some discussion to address this concern before the projected application.

In summary, I would suggest the author revise the current version before a resubmission.

Review form: Reviewer 3

Is the manuscript scientifically sound in its present form?

Yes

Are the interpretations and conclusions justified by the results?

Yes

Is the language acceptable?

Yes

Do you have any ethical concerns with this paper?

Yes

Have you any concerns about statistical analyses in this paper?

No

Recommendation?

Major revision is needed (please make suggestions in comments)

Comments to the Author(s)

The manuscript describes the photothermal mediated antibacterial efficacy of GO-Ag nanocomposites against clinical isolated MDR E. coli. I have some concerns related to the manuscript. After addressing the following concerns, the manuscript could be suitable for publication in 'Royal Society Open Science'.

Some specific concerns:

i) The following title of the manuscript would be more appropriate.

'Photothermal-assisted antibacterial application of GO-Ag nanocomposites against clinical isolated MDR E. coli'

ii) 'GO-Ag nanocomposites showed the highest antibacterial efficiency among tested antimicrobials...'

The tested antimicrobials should be mentioned in the Abstract.

iii) 'The remained bacteria viabilities were 4.4% and 4.1% respectively for different bacteria strains with...'

The different bacteria strains should be mentioned in the Abstract.

- iv) Full form of MIC should be given when it is first time used in Introduction.
- v) Full form of MTT should be given when it is first time used in Introduction.
- vi) 'Finally, characterization of GO-Ag treated bacteria is performed using confocal microscopy and...'
It seems from the result that 'fluorescence microscopy' was used instead of 'confocal microscopy'.
- vii) Figure 4: It looks like the fluorescence intensity in E. Coli for Go-Ag nanocomposites in absence and presence of laser light is almost similar. How can you explain this result in terms of better therapeutic efficacy of the nanocomposites in presence of laser light as compared to only nanocomposites without laser light?
- viii) Figure 1d: Why did you take different concentration of AgNP, GO and GO-Ag for photothermal experiments? It should be same to compare the photothermal efficacy of the particular material.
- ix) How did you measure the content of silver in AgNPs and GO-Ag nanocomposites using inductively coupled plasma atomic emission spectroscopy (ICP-AES)? You should elaborate the result and experimentation.
- x) Figure 2, Figure 3b: Statistical significance with p value should be provided.

Decision letter (RSOS-192019.R0)

05-Mar-2020

Dear Dr Wang:

Title: Photothermal-assisted antibacterial application of GO-Ag against clinical isolated MDR E. coli

Manuscript ID: RSOS-192019

The editor assigned to your manuscript has now received comments from reviewers. We would like you to revise your paper in accordance with the referee and Subject Editor suggestions which can be found below (not including confidential reports to the Editor). Please note this decision does not guarantee eventual acceptance.

Please submit your revised paper before 28-Mar-2020. Please note that the revision deadline will expire at 00.00am on this date. If we do not hear from you within this time then it will be assumed that the paper has been withdrawn. In exceptional circumstances, extensions may be possible if agreed with the Editorial Office in advance. We do not allow multiple rounds of revision so we urge you to make every effort to fully address all of the comments at this stage. If deemed necessary by the Editors, your manuscript will be sent back to one or more of the original reviewers for assessment. If the original reviewers are not available we may invite new reviewers.

To revise your manuscript, log into <http://mc.manuscriptcentral.com/rsos> and enter your Author Centre, where you will find your manuscript title listed under "Manuscripts with Decisions." Under "Actions," click on "Create a Revision." Your manuscript number has been

appended to denote a revision. Revise your manuscript and upload a new version through your Author Centre.

RSC Associate Editor:
Comments to the Author:
(There are no comments.)

RSC Subject Editor:
Comments to the Author:
(There are no comments.)

Reviewers' Comments to Author:
Reviewer: 1

Comments to the Author(s)
Remarks to Authors,

This manuscript describes a graphene oxide-silver (GO-Ag) nanocomposites as an antibacterial agent to treat multidrug-resistant (MDR) bacterial infection. Ag nanoparticles have an excellent antibacterial effect due to the toxicity of Ag ion. GO mediated photothermal effect will potentially cause bacterial death. Synergetic antibacterial effects can against clinical isolated MDR bacterial. However, some major defects make it difficult to accept.

1. The characterization of GO-Ag nanocomposites is not clear. The contents of each component are unknown, such as the ratio of GO and Ag in GO-Ag, respectively. It is not simply put these two together, which may cause wrong results. Such as the concentration of the three components Ag NP, GO, GO-Ag were 4 $\mu\text{g m/L}$, 3 $\mu\text{g m/L}$ and 7 $\mu\text{g m/L}$. And It also may cause some unmatched results such as in Figures 1c and 1d. The absorbance of GO-Ag at 808 nm is lower than the GO, however, the photothermal effect of GO-Ag is higher than the GO at the same

- concentration of GO. These results indicated that the concentration of GO-Ag may be wrong. The more detail quantitatively analysis of the GO-Ag is needed, such as through the TGA and ICP-MS.
2. In Figure 3b, the GO mediated photothermal effect shows a dramatically antibacterial efficiency at a power density of 2 W/cm². however, it cannot be observed at a power density of 1.5 W/cm². The results should be carefully checked again. And the antibacterial transition power density between 2 and 1.5 W/cm² should be provided to make it more convincing.
 3. The results in Figure 4 (Laser- GO-Ag+ group) shows that less than 20 % of cells are alive, which are not consistent with the results in Figure 2 (at least 60 % cells alive at the same concentration of GO-Ag).
 4. The English writing should be much improved.

Reviewer: 2

Comments to the Author(s)

The authors report antibacterial and photothermal applications of GO-Ag nanocomposite using clinical isolated bacteria as examples. Nevertheless, there are some concerns as below:

1) Characterization of GO-Ag / quality control

To reproducibly obtain the same results, proper physicochemical characterization of the GO-Ag nanocomposite should be done. E.g. what about the zeta-potential of the prepared GO-Ag nanocomposites, which actually affect their interaction with bacterial membrane? Is the GO single layer or multiple layers? What about the size distribution of Ag NP synthesized on GO?

2) Toxicity of GO-Ag

When the author talk about future medical applications in section "5.Discussion", they claim that "...GO-Ag nanocomposites can be coated onto consumables such as bandage, gauze and woundplast, or surgical devices...". However, there is a serious concern on the toxicity of Ag/Ag⁺ to human body? There should be some discussion to address this concern before the projected application.

In summary, I would suggest the author revise the current version before a resubmission.

Reviewer: 3

Comments to the Author(s)

The manuscript describes the photothermal mediated antibacterial efficacy of GO-Ag nanocomposites against clinical isolated MDR E. coli. I have some concerns related to the manuscript. After addressing the following concerns, the manuscript could be suitable for publication in 'Royal Society Open Science'.

Some specific concerns:

i) The following title of the manuscript would be more appropriate.

'Photothermal-assisted antibacterial application of GO-Ag nanocomposites against clinical isolated MDR E. coli'

ii) 'GO-Ag nanocomposites showed the highest antibacterial efficiency among tested antimicrobials...'

The tested antimicrobials should be mentioned in the Abstract.

iii) 'The remained bacteria viabilities were 4.4% and 4.1% respectively for different bacteria strains with...'

The different bacteria strains should be mentioned in the Abstract.

iv) Full form of MIC should be given when it is first time used in Introduction.

v) Full form of MTT should be given when it is first time used in Introduction.

vi) 'Finally, characterization of GO-Ag treated bacteria is performed using confocal microscopy and...'

It seems from the result that 'fluorescence microscopy' was used instead of 'confocal microscopy'.

vii) Figure 4: It looks like the fluorescence intensity in E. Coli for Go-Ag nanocomposites in absence and presence of laser light is almost similar. How can you explain this result in terms of better therapeutic efficacy of the nanocomposites in presence of laser light as compared to only nanocomposites without laser light?

viii) Figure 1d: Why did you take different concentration of AgNP, GO and GO-Ag for photothermal experiments? It should be same to compare the photothermal efficacy of the particular material.

ix) How did you measure the content of silver in AgNPs and GO-Ag nanocomposites using inductively coupled plasma atomic emission spectroscopy (ICP-AES)? You should elaborate the result and experimentation.

x) Figure 2, Figure 3b: Statistical significance with p value should be provided.

Author's Response to Decision Letter for (RSOS-192019.R0)

See Appendix A.

RSOS-192019.R1 (Revision)

Review form: Reviewer 3

Is the manuscript scientifically sound in its present form?

Yes

Are the interpretations and conclusions justified by the results?

Yes

Is the language acceptable?

Yes

Do you have any ethical concerns with this paper?

No

Have you any concerns about statistical analyses in this paper?

No

Recommendation?

Accept as is

Comments to the Author(s)

The authors have tried their best to address all the concerns and significantly improved the quality of the manuscript. Therefore, the manuscript is now suitable for publication in 'Royal Society Open Science'.

Decision letter (RSOS-192019.R1)

Dear Dr Wang:

Title: Photothermal-assisted antibacterial application of GO-Ag nanocomposites against clinical isolated MDR E. coli

Manuscript ID: RSOS-192019.R1

It is a pleasure to accept your manuscript in its current form for publication in Royal Society Open Science. The chemistry content of Royal Society Open Science is published in collaboration with the Royal Society of Chemistry.

RSC Associate Editor:
Comments to the Author:
(There are no comments.)

RSC Subject Editor:
Comments to the Author:
(There are no comments.)

Reviewer(s)' Comments to Author:

Reviewer: 3

Comments to the Author(s)

The authors have tried their best to address all the concerns and significantly improved the quality of the manuscript. Therefore, the manuscript is now suitable for publication in 'Royal Society Open Science'.

Appendix A

Reviewers' Comments to Author:

Reviewer: 1

Comments to the Author(s)

Remarks to Authors ,

This manuscript describes a graphene oxide-silver (GO-Ag) nanocomposites as an antibacterial agent to treat multidrug-resistant (MDR) bacterial infection. Ag nanoparticles have an excellent antibacterial effect due to the toxicity of Ag ion. GO mediated photothermal effect will potentially cause bacterial death. Synergetic antibacterial effects can against clinical isolated MDR bacterial. However, some major defects make it difficult to accept.

1. The characterization of GO-Ag nanocomposites is not clear. The contents of each component are unknown, such as the ratio of GO and Ag in GO-Ag, respectively. It is not simply put these two together, which may cause wrong results. Such as the concentration of the three components Ag NP, GO, GO-Ag were 4 $\mu\text{g m/L}$, 3 $\mu\text{g m/L}$ and 7 $\mu\text{g m/L}$. And It also may cause some unmatched results such as in Figures 1c and 1d. The absorbance of GO-Ag at 808 nm is lower than the GO, however, the photothermal effect of GO-Ag is higher than the GO at the same concentration of GO. These results indicated that the concentration of GO-Ag may be wrong. The more detail quantitatively analysis of the GO-Ag is needed, such as through the TGA and ICP-MS.

Reply: Actually, silver concentration in GO-Ag nanocomposites was measured by ICP-AES after being decomposed by nitric acid. The ratio GO and Ag in GO-Ag was mentioned in section "4.3 Synergetic antibacterial effect verification of GO-Ag", "Results showed content of GO and AgNP were 57% and 43% respectively in GO-Ag nanocomposite." As a result, 4 $\mu\text{g/mL}$ AgNP and 3 $\mu\text{g/mL}$ GO were used as control group for comparison antibacterial study. However, in order to minimize misunderstandings, component content of GO-Ag nanocomposites will be added in Table 1 and described in section "4.1 Antibacterial agents' characterization" in revised version.

Nanomaterials used in Figure 1c and Figure 1d are not the same concentration. Specifically, in Figure 1c, GO concentration in GO-Ag nanocomposites is not the same with GO nanomaterial. Figure 1c will be updated using exactly same concentration nanomaterial in Figure 1d. Moreover, solvent used in Figure 1c and Figure 1d was not the same. Distilled water and LB broth were used in Figure 1c and Figure 1d respectively. LB broth is a nutritionally rich medium used for cultivation of *E. coli*. Its formulation is an industry standard, which contains 1% tryptone, 0.5% yeast extract and 1% NaCl. Clear legend will be updated to eliminate misunderstandings.

Revision:

- 1) Ag and GO content in GO-Ag was added in Table 1 in revised version.

Table 1. Component content of GO-Ag and AgNP

Sample	Dilution Factor	Ag Content ($\mu\text{g/mL}$)		GO Content ($\mu\text{g/mL}$)
		AgNP	GO-Ag	GO-Ag
#1	2	462	188	267
#2	5	193	91	108
#3	10	104	38	53
Mean Dose ($\mu\text{g/mL}$)		976	404	535
Content (%)			43%	57%

Ag content was quantified using ICP-AES, and GO content was measured using spectrometer at 900 nm (extinction coefficient was $21.2 \text{ mg mL}^{-1} \text{ cm}^{-1}$). Dose = dilution factor * measured content.

2) Absorption spectrum in Figure 1 was updated.

2. In Figure 3b, the GO mediated photothermal effect shows a dramatically antibacterial efficiency at a power density of 2 W/cm^2 . however, it cannot be observed at a power density of 1.5 W/cm^2 . The results should be carefully checked again. And the antibacterial transition power density between 2 and 1.5 W/cm^2 should be provided to make it more convincing.

Reply : Thanks for your suggestions. We strongly agree that any unusual data should be checked carefully. In this experiment, 3 parallel samples were photothermal treated before testing, 3 replicates were tested in MTT assay for each sample. As a result, we have 9 data results for each column in Figure 3b. Besides, we have evidence that results are reliable after data comparison between GO and GO-Ag.

As shown in heating curves (Figure S3) at different power density. Heating curve of 1 W/cm^2 -GO-Ag is similar with 1.5 W/cm^2 -GO, and 1.5 W/cm^2 -GO-Ag is similar with 2.0 W/cm^2 -GO. As

shown in Figure 2b, photothermal effect of GO was the only factor that contribute to antibacterial effect, similar heating curve means similar antibacterial effect.

Results from Figure 3 shows that viability decrement of GO from 0 W/cm² to 1.5 W/cm² is 10.8% while GO-Ag from 0 W/cm² to 1.0 W/cm² is 17.0%. Remained Viabilities of 2.0 W/cm² GO group and 1.5 W/cm² GO-Ag group are 6% and 3% respectively. We can observe similar antibacterial effect after data analysis. So, we believe our results are reliable.

Revision: Heating curves of GO-Ag and GO was provided in Figure S3 in order to make it more convincing.

Figure S3. Heating curves of GO and GO-Ag at different power density.

808 nm laser irradiation was conducted on GO and GO-Ag dispersed in LB broth at room temperature. Concentration of GO and GO-Ag were 3 μg mL⁻¹ and 7 μg mL⁻¹ respectively, and temperature was recorded by thermal camera.

3. The results in Figure 4 (Laser- GO-Ag+ group) shows that less than 20 % of cells are alive, which are not consistent with the results in Figure 2 (at least 60 % cells alive at the same concentration of GO-Ag).

Reply: It is well known that fluorescent imaging is a qualitative method to identify cell living status. Compared with million to billion bacteria cells used to quantify cell variability in Figure 2, the sample size in Figure 4 is extremely small. In the perspective of statistics, we do not believe images in Figure 4 are suitable for cell viability assessment. The purpose of Figure 4 is to identify living status of bacteria in different treatment groups using a well-accepted method. However, we really appreciate the reviewer's remarks, which is very helpful to fully understand the data we collected in another way.

Revision: Revision was made in section '4.5 Characterization of GO-Ag treated MDR bacteria' in order to clarify the purpose and correct improper description of the results.

'Fluorescence imaging, a well-accepted qualitative method, was used to distinguish living and non-living bacteria in treated groups (red color in Figure 4) other than efficiency comparison (Figure 3). Specifically, compared with million to billion cells used in viability assessment in

Figure 3, fluorescence images were snap shots of extremely small parts of treated bacteria, which was not suitable for quantitative comparisons.'

4. The English writing should be much improved.

Reply: Yes, we will do our best to improve English writing.

Reviewer: 2

Comments to the Author(s)

The authors report antibacterial and photothermal applications of GO-Ag nanocomposite using clinical isolated bacteria as examples. Nevertheless, there are some concerns as below:

1)Characterization of GO-Ag / quality control

To reproducibly obtain the same results, proper physicochemical characterization of the GO-Ag nanocomposite should be done. E.g. what about the zeta-potential of the prepared GO-Ag nanocomposites, which actually affect their interaction with bacterial membrane? Is the GO single layer or multiple layers? What about the size distribution of Ag NP synthesized on GO?

Reply: Thanks for your important suggestions. More characterization data of GO-Ag will be added to make it reproducible.

Revision:

Figure 1 was updated with AFM analysis of GO, size distribution of AgNP on GO, and zeta potential of GO-Ag. Besides, zeta potential of GO and size distribution of GO-Ag were also updated in Figure S1 and Figure S2 as supporting information.

Results were described in section "4.1 Antibacterial agents' characterization". Zeta-potential of GO-Ag was -28.8 ± 4.51 mV. Both single layer and multilayer GO were exist in synthesized GO. Average diameter of AgNP on GO was 24.38 ± 10.74 nm.

Figure 1. Characterization of synthesized nanomaterials.

(a) TEM image of GO. (b) AFM image of GO. (c) TEM image of GO-Ag nanocomposites. (d) size distribution of AgNP on GO-Ag. (e) Zeta potential distribution of GO-Ag. (f) Absorption spectrum of GO, AgNP and GO-Ag dispersed in water.

Figure S1. Zeta potential distribution of GO dispersed in water.

Zeta potential was measured using instrument named Zetasizer Nano (ZS90, Malvern). Zeta potential of GO was -36.1 ± 6.98 mV.

Figure S2. Size distribution of GO-Ag nanocomposites dispersed in water.

GO-Ag nanocomposites diameter were measured using Dynamic Light Scattering (DLS) method. Diameter of GO-Ag was 147.4 ± 85.16 nm.

2) Toxicity of GO-Ag

When the author talk about future medical applications in section "5.Discussion", they claim that "...GO-Ag nanocomposites can be coated onto consumables such as bandage, gauze and woundplast, or surgical devices...". However, there is a serious concern on the toxicity of Ag/Ag⁺ to human body? There should be some discussion to address this concern before the projected application.

In summary, I would suggest the author revise the current version before a resubmission.

Reply : Thanks for your advice. Toxicity of synthesized nanomaterials are always the concerns before projected application. Cell toxicity data of GO-Ag nanocomposite and corresponding toxicity discussion will be added in "Discussion" section.

Revision:

Toxicity of GO-Ag was evaluated using human derived cell lines. Results was shown in Figure 6.

"...GO-Ag nanocomposites can be coated onto consumables such as bandage, gauze and woundplast, or surgical devices..." was deleted as no direct evidences can fully address serious concerns on toxicity to human bodies at present. As recommended by the reviewer, any projected applications should be carefully proved.

Figure 6. Cytotoxicity of GO-Ag nanocomposites

HEK 293T and Hep G2 cells viability assessment was conducted using MTT assay after 24 hours incubation with GO-Ag nanocomposites. Error bars are standard deviations of 5 replicates.

* $p < 0.05$, ** $p < 0.01$, *** $p < 0.001$.

Reviewer: 3

Comments to the Author(s)

The manuscript describes the photothermal mediated antibacterial efficacy of GO-Ag nanocomposites against clinical isolated MDR E. coli. I have some concerns related to the

manuscript. After addressing the following concerns, the manuscript could be suitable for publication in 'Royal Society Open Science'.

Some specific concerns:

i) The following title of the manuscript would be more appropriate.

'Photothermal-assisted antibacterial application of GO-Ag nanocomposites against clinical isolated MDR E. coli'

Revision: Thanks for your help. Title will be revised according to your suggestion.

ii) 'GO-Ag nanocomposites showed the highest antibacterial efficiency among tested antimicrobials...'

The tested antimicrobials should be mentioned in the Abstract.

Revision: Revised description is 'GO-Ag nanocomposites showed the highest antibacterial efficiency among tested antimicrobials (graphene oxide, silver nanoparticles, GO-Ag)'

iii) 'The remained bacteria viabilities were 4.4% and 4.1% respectively for different bacteria strains with....'

The different bacteria strains should be mentioned in the Abstract.

Revision: Revised sentence will be '14.0 $\mu\text{g mL}^{-1}$ GO-Ag treatment could greatly inhibit bacteria growth, remained bacteria viabilities were 4.4% and 4.1% for MDR-1 and MDR-2 E. coli bacteria respectively.'

iv) Full form of MIC should be given when it is first time used in Introduction.

Revision: We will add full name of MIC (minimum inhibitory concentration) in introduction section. Revised sentence was 'Minimum inhibitory concentration (MIC) of AgNPs, GO-Ag, and some antibiotics...'

v) Full form of MTT should be given when it is first time used in Introduction.

Reply: Thanks for your help. We will check the whole manuscript and revise similar mistakes.

Revision: '3-(4,5-dimethyl-2-thiazolyl)-2,5-diphenyl-2H-tetrazolium bromide', full name of MTT will be added in Introduction. Revised sentence was '...are studied simultaneously using a colorimetric assay with dye 3-(4,5-dimethyl-2-thiazolyl)-2,5-diphenyl-2H-tetrazolium bromide (MTT), namely MTT assay.'

vi) 'Finally, characterization of GO-Ag treated bacteria is performed using confocal microscopy and...'

It seems from the result that 'fluorescence microscopy' was used instead of 'confocal microscopy'.

Reply: Thanks for your help. 'confocal microscope' is the instrument used for fluorescence imaging, 'fluorescence microscopy' is the name of imaging technique. We will check the whole manuscript and revise mistakes.

Revision: 'confocal microscopy' was replaced with 'fluorescence microscopy' in Introduction section.

vii) Figure 4: It looks like the fluorescence intensity in *E. Coli* for Go-Ag nanocomposites in absence and presence of laser light is almost similar. How can you explain this result in terms of better therapeutic efficacy of the nanocomposites in presence of laser light as compared to only nanocomposites without laser light?

Reply: We can't fully understand this question. Suppose your point is similar fluorescence intensity (red light) means similar therapeutic efficacy. In our view, more red light means more dye stained with nucleic acid in *E. coli*. Once cell membrane integrity was damaged, propidium iodide (PI) will enter into cell and bind with nucleic acid with little or no sequence preference. So, we believe there is no obvious fluorescence intensity difference once membrane integrity was damaged. Cell membrane integrity disruption was observed by SEM after GO-Ag treatment.

In Figure 4, GO-Ag treated groups, total cell number of photothermal group is much less than absence group. However, ratio of non-living bacteria might not always consistent with quantitative results in Figure 3d because sample size in Figure 4 is much smaller than quantitative method. Figure 4 provides direct evidence that bacteria were non-living after GO-Ag treatment in presence and absence of laser light.

viii) Figure 1d: Why did you take different concentration of AgNP, GO and GO-Ag for photothermal experiments? It should be same to compare the photothermal efficacy of the particular material.

Reply: It's a good point. The intended purpose is not for photothermal efficacy characterization. We wanted to provide heating curves of materials used in photothermal experiment in Figure 3. For this purpose, Figure 1d should be combined in photothermal treatment section.

Revision: Figure 1d was moved to Figure 3 for better organization of this manuscript.

Figure 3. In-vitro photothermal treatment of MDR-2 E. coli.

ix) How did you measure the content of silver in AgNPs and GO-Ag nanocomposites using inductively coupled plasma atomic emission spectroscopy (ICP-AES)? You should elaborate the result and experimentation.

Reply: We will provide details of silver content measurement in ‘Materials and Methods’ section, and elaborate results in Table 1.

Revision:

Table 1. Component content of GO-Ag and AgNP

Sample	Dilution Factor	Ag Content ($\mu\text{g}/\text{mL}$)		GO Content ($\mu\text{g}/\text{mL}$)
		AgNP	GO-Ag	GO-Ag
#1	2	462	188	267
#2	5	193	91	108
#3	10	104	38	53
Mean Dose ($\mu\text{g}/\text{mL}$)		976	404	535
Content (%)			43%	57%

Ag content was quantified using ICP-AES, and GO content was measured using spectrometer at 900 nm (extinction coefficient was $21.2 \text{ mg mL}^{-1} \text{ cm}^{-1}$). Dose = dilution factor * measured content.

x) Figure 2, Figure 3b: Statistical significance with p value should be provided.

Response: It’s a very good suggestion, p value is widely used for statistical significance judgement. p value will be added in Figure 2 and Figure 3b.

Revision: revised Figure will be updated in revision version as shown below.

Figure 2. MDR E. coli viability assessment by MTT assay.

Figure 3. In-vitro photothermal treatment of MDR-2 E. coli.

We thank all reviewers for their insightful comments and suggestions, which greatly helped to improve the quality of our manuscript!